# The Quality, Reliability, and Accuracy of Videos Regarding Exercises and Management for Dysphagia in Pediatric Populations Uploaded on YouTube

**DOI:** 10.3390/children9101514

**Published:** 2022-10-04

**Authors:** Min Cheol Chang, Byung Joo Lee, Donghwi Park

**Affiliations:** 1Department of Rehabilitation Medicine, Yeungnam University Hospital, Daegu 41404, Korea; 2Department of Rehabilitation Medicine, Daegu Fatima Hospital, Daegu 41199, Korea; 3Department of Physical Medicine and Rehabilitation, College of Medicine, Ulsan University Hospital, University of Ulsan, Ulsan 44033, Korea

**Keywords:** YouTube, video, dysphagia, pediatric, exercise, management, quality

## Abstract

**Objectives:** YouTube is well known for providing easy access to various kinds of video content. In this study, we investigated the quality and reliability of videos on YouTube addressing exercise or dysphagia management in the pediatric population. **Methods:** Video quality and reliability were assessed by using the Global Quality Scale (GQS) and a modified DISCERN tool, respectively. The accuracy of the information in each video was also evaluated. Other information, including the video source, length, date of upload, as well as the number of views, likes, dislikes, and comments were investigated, and statistical significance was determined. **Results:** In total, 22 videos on exercises and dysphagia management in pediatric populations were evaluated; 36.4% and 72.7% of these videos did not have high quality or reliability, respectively. Moreover, half of the videos did not contain accurate information. Even when videos were created by medical specialists, many of these YouTube videos were of low quality, reliability, and accuracy. **Conclusions:** The reliability, quality, and accuracy of many videos on exercise or dysphagia management in the pediatric population were low. Video creators, especially medical specialists, should strive to create videos with high quality, reliability, and accuracy.

## 1. Introduction

The importance of swallowing difficulties in the pediatric population is increasing with time and awareness. Up to 50% of parents report that their otherwise healthy children have at least one problem with feeding [1]. Dysphagia is becoming increasingly common in the pediatric population, especially as advances in healthcare management improve the survival of extremely premature infants and children with neurological disorders or complex congenital anomalies [2,3].

Dysphagia in the pediatric population can result from a variety of causes, including prematurity, physiological and anatomical disorders, and neurological impairment occurring anywhere between the mouth and esophagus [2]. Various management approaches have been reported to improve or treat the symptoms of dysphagia in the pediatric population [2,3]. Feeding therapy, performed by an occupational therapist or an experienced speech and language pathologist, is often the first-line treatment for pediatric patients with swallowing difficulty [4,5]. This therapy may include altering the feeding position, changing the means of food delivery, including nipple flow, milk bottle, or spoon, or adjusting the speed of feeds to improve the suck–swallow–breathe pattern [6]. Motor exercise and sensory stimulation performed by a trained speech and language pathologist or occupational therapist also improve movement, strength, and coordination of the tongue, lips, hard and soft palates, and laryngeal and pharyngeal muscles [6].

Recently, the internet has expanded as a primary means of disseminating information and its beneficial effects around the world [7], and obtaining health-related information from the Internet is gradually becoming common. In the previous research, it was reported that as much as 50% of the adult population in the United States get health-related informations by searching the Internet [8,9]. As a common video-sharing website, YouTube is now widely used to make videos available to users [10,11]. Due to its extensive character and free video content, the YouTube website can be considered a useful tool for finding and sharing health-related information [12]. Consequently, it can be also considered a useful tool for patient education.

However, there are worries regarding the quality, reliability, and accuracy of YouTube videos. Given the characteristics of videos on YouTube, which allows any person to upload video without verification and which can also be used for advertising purposes, the reliability, quality, and accuracy of the uploaded videos must be verified [13]. To date, no studies have examined the quality, reliability, and accuracy of YouTube videos related to dysphagia exercises and management in pediatric populations.

Here, we investigated the quality, reliability, and accuracy of YouTube videos on exercises and management for dysphagia in pediatric populations.

## 2. Methods

### 2.1. Video Selection

When searching for videos on YouTube, the keywords “pediatric dysphagia rehabilitation,” “pediatric dysphagia exercise,” “pediatric dysphagia therapy,” “pediatric dysphagia management,” and “pediatric dysphagia treatment” were used in this study (http://www.youtube.com, accessed on 19 August 2022). Keywords for 60 videos in total (the first three pages) of YouTube videos in the English language were separately analyzed by two physiatrists with >12 years of experience in the management and treatment of dysphagia. A previous study has claimed that most YouTube viewers watch videos on the first three pages of a query result [10,11,14]. Therefore, we assumed that evaluating the first three pages would adequately represent the majority of YouTube users. YouTube videos were analyzed based on the number of views, resulting in the most-viewed videos being shown at the top. Two expert physiatrists (B.L and D.P) assessed 200 videos. Duplicated videos, off-topic videos, videos with inappropriate sound quality (e.g., inaudible), and videos in languages other than English were excluded from this study. Two reviewers (B.L. and D.P.) assessed the quality, reliability, and accuracy of the included videos, and the discrepancies in the assessment were discussed to reach a consensus (Figure 1). A list of YouTube video links is provided in Appendix A.

### 2.2. Assessment of Quality, Reliability, and Accuracy

As a tool designed to assess the quality of Internet resources [15], the Global Quality Scale (GQS) was utilized by two independent expert physiatrists (B.L. and D.P.) to investigate the educational qualities of YouTube videos (Table 1). The GQS consists of a five-point scale, ranging from 1 to 5 [15], wherein a higher GQS score denotes a higher quality of information. The researchers investigate the quality, flow, and use of the videos by using this scale. The video was considered to be of high quality if it obtained a score of 4–5 points, intermediate quality if it obtained 3 points, and low quality if it obtains 1 or 2 points. A consensus was made by discussion if there was disagreement regarding the scores for the same videos.

The modified DISCERN scale (mDISCERN) [16] was used to investigate the reliability of the YouTube videos. The mDISCERN involves five questions, and each question is answered with either a “yes” or a ”no” [16,17], with each yes receiving one point (Table 2) [17]. A higher mDISCERN score indicates greater reliability. When the mDISCERN score is ≥3, the information is considered highly reliable.

All videos were categorized as misleading, accurate, or neither misleading nor accurate. If the videos contained at least one inaccurate or correct scientific information about exercises for and the management of pediatric dysphagia, they were categorized as misleading videos and accurate videos, respectively. Videos without any such pertinent information were considered neither accurate nor misleading, and those that contained both inaccurate and accurate information were categorized as misleading. The accuracy of the included videos was evaluated based on a review article written by Umay et al. [6]. This previous study, using a seven-step process and a third-modified Delphi survey, provided comprehensive and detailed answers and recommendations to all issues that may arise in clinical practice for the management of dysphagia in children from the perspective of an experienced multidisciplinary expert.

### 2.3. Data Extraction

Data on the length of the video, upload date, as well as the number of views, likes, dislikes, and comments were collected for each video [18]. The total number of views, likes, dislikes, and comments was divided by the number of days since the video had been uploaded to YouTube [18] to calculate the total number of likes, dislikes, and views, comments and views per day.

### 2.4. Sources of the Videos

The sources of the videos were classified into eight categories: (1) therapists (occupational and speech), (2) physicians, (3) health-related websites, (4) academies, (5) associations, professional organizations, or universities, (6) non-physician health people, (7) patients, and (8) independent users [18].

### 2.5. Statistical Analysis

The GQS, mDISCERN scores, upload date, and number of views, likes, dislikes, and comments are presented as mean scores ± standard mean values. The rates (%) of classification of quality, reliability, and accuracy were calculated according to the source of the videos.

A Pearson’s correlation test was used to identify the correlations between quality, reliability, accuracy, and other video parameters. By using only the parameters that were significant in this test, multiple linear regression analyses were performed. Multivariate regression analyses with stepwise methods were performed. Statistical analyses were performed by using the Statistical Package for the Social Sciences for Windows and the R package for Windows (version 2.15.2; R Foundation for Statistical Computing, Vienna, Austria) with the statistical significance set at *p* < 0.05.

### 2.6. Ethics Statement

This study did not involve any human or animal participants. This study included publicly accessible YouTube videos. Approval from the ethics committee was, therefore, not required for this study.

## 3. Results

The study excluded 91 off-topic videos, 115 duplicate videos, 70 videos in languages other than English, and 2 videos with unsuitable audio from a total of 300 videos. A total of 22 videos were investigated. The characteristics of the videos on YouTube, such as the length of the video, and the number of views, likes, dislikes, and comments, are summarized in Table 3.

Among these, 9 (40.8%) videos were uploaded by therapists, 4 (18.2%) by physicians or physiatrists, 1 (4.6%) by a health-related website, 3 (13.6%) by a university/professional organization/association, 1 (4.6%) by a non-physician health worker, and 4 (18.2%) by independent users.

Based on the assessment of information quality, the mean GQS score of the 22 included videos was 3.68 ± 0.95, implying an intermediate level of quality on average. According to the GQS scores, among the 22 videos, 14 (63.6%), 6 (27.3%), and 2 (9.1%) were of high, intermediate, and low quality, respectively (Table 4). Based on the sources of the YouTube videos, 6 (66.7%) of the 9 videos made by therapists, 3 (75.0%) of the 4 videos made by physicians or physiatrists, 2 (66.7%) of the 3 videos made by universities/professional organizations/associations, 2 (50.0%) of the 4 videos made by independent users, and 1 (100%) video made by non-physician health personnel were assessed to be of high quality (Table 4).

As for the reliability of the videos, the mean mDISCERN score of the included 22 videos was 1.77 ± 1.34, which implied non-reliability on average. Among the 22 videos, 6 (27.3%) were reliable (mDISCERN score ≥ 3) and 16 (62.7%) were non-reliable (mDISCERN score < 3) (Table 5). Based on the sources of the YouTube videos, 3 (33.3%) of the 9 videos made by therapists, 1 (25.0%) of the 4 videos made by physicians or physiatrists, 2 (66.7%) of the 3 videos made by universities/professional organizations/associations, 0 (0.0%) of the 1 video made by non-physician health personnel, 0 (0.0%) of the 4 videos made by independent users, and 1 (100%) video made by non-physician health personnel were assessed to be reliable based on the mDISCERN scores (Table 5).

Regarding the accuracy of the videos, 11 (50.0%) of the 22 videos were accurate, whereas the remaining 11 (50.0%) videos were misleading (Table 6). According to the sources of the YouTube videos, 5 (55.6%) of the 9 videos made by therapists, 3 (75.0%) of the 4 videos made by physicians or physiatrists, 2 (66.7%) of the 3 videos made by universities/professional organizations/associations, 0 (0.0%) of the 1 video made by non-physician health personnel, and 1 (25.0%) of the 4 videos made by independent users were assessed to be accurate (Table 6).

Pearson’s correlation between the GQS, mDISCERN scores, accuracy, and other video parameters revealed significant correlations between the GQS and mDISCERN scores, between the GQS and accuracy, between the mDISCERN scores and accuracy, and between the GQS and views per day (Table 7). However, there were significant positive correlations between the GQS and mDISCERN scores and between the GQS and accuracy (*p* < 0.05) in the multivariate regression analysis (Table 8).

## 4. Discussion

According to the results of this study, 14 (63.6%), 6 (27.3%), and 2 (9.1%) of the 22 videos evaluated were of high, intermediate, and low quality based on GQS, respectively. Although 63.6% of videos on YouTube about exercises and the management of pediatric dysphagia were of high quality, 36.4% were intermediate and low-quality videos. According to the sources of the YouTube videos, 9 of the 13 videos made by medical specialists (6 of the 9 videos by therapists, and 3 of the 4 videos by physicians or physiatrists) were of high quality. This implies that YouTube videos uploaded by medical professionals, such as physicians/physiatrists or therapists, may not always be of high quality.

Regarding reliability, the mean mDISCERN score of the 22 videos included was 1.77 ± 1.34, indicating that the videos uploaded to YouTube on exercises for and the management of pediatric dysphagia were not reliable on average. Based on mDISCERN scores, 6 (27.3%) of the 22 videos were reliable, whereas the remaining 16 (62.7%) videos were not. According to the sources of the YouTube video, only 4 (30.77%) of the 13 videos made by medical specialists (3 of the 9 videos made by therapists, and 1 of the 4 videos made by physicians or physiatrists) were reliable based on mDISCERN scores. This indicated that many videos uploaded on YouTube on exercises for and the management of pediatric dysphagia were not reliable, even when uploaded by medical professionals.

As for accuracy, 11 (50.0%) of the 22 videos were accurate, whereas the remaining 11 (50.0%) videos were misleading. According to the sources of the YouTube videos, 8 (61.54%) of the 13 videos made by medical specialists (5 of the 9 videos by therapists, and 3 of the 4 videos by physicians or physiatrists) were accurate. This implied that half of the videos uploaded to YouTube on exercises for and the management of pediatric dysphagia were inaccurate, even when uploaded by medical specialists. Our results indicate that videos uploaded on this website can be of low quality, reliability, and accuracy even when uploaded by medical professionals. Medical specialists should review previous literature thoroughly before creating their videos and making their videos based on verified and accurate facts, and non-medical professionals also need to consult medical specialists who have enough knowledge on the topic. Additionally, in our study, there were no YouTube videos uploaded by health organizations and/or medical associations previously reported as the most reliable sources. This is thought to be due to the characteristics of the subjects we investigated [13].

We also investigated the correlations among quality, reliability accuracy, and other video parameters of YouTube videos on the topic of exercises for and the management of pediatric dysphagia and found a significant and positive correlation between video quality and reliability and between video quality and accuracy. For creating high-quality videos, creators should make their videos accurately while referring to reliable sources.

So far, several studies assessing the quality of YouTube videos were conducted on various medical topics. Videos on rheumatoid arthritis, dialysis, and self-injection of anti-tumor necrosis factor agents showed similar results to those of our study (63.6%, high-quality videos; 50%, accurate videos) [19,20,21] and that up to 50% of these YouTube videos included useful medical information and good quality. However, an analysis of videos on pediatric retinopathy, methotrexate self-injection, and male urethral catheterization showed that only 2–20% of them were useful or highly qualified [14,22,23]. The quality of YouTube videos on the topic of exercises and management for pediatric dysphagia is comparatively relatively high. This might be because dysphagia is a topic about which the public is less knowledgeable, making it difficult to upload videos about it and leading to most videos on the topic being uploaded by physicians, therapists, and medical institutions.

This study had a few limitations. First, our tools for the assessment of the quality, reliability, and accuracy of videos are subjective and can be easily affected by bias. For this reason, the results of this study may not be reproduced when re-analyzed by other researchers. Secondly, the number of assessed videos was relatively small. Therefore, the results of our study may not represent the characteristics of all YouTube videos on the topic of exercises and management for pediatric dysphagia. In addition, because we investigated only at a specific point in time, our research results may not be reproduced due to the nature of the YouTube website on which new videos are continuously uploaded. Considering the characteristics of this YouTube website, it may also be interesting to re-analyze the above topic after a few years. Further studies compensating for these limitations are required in the future.

## 5. Conclusions

In this study, we investigated the quality, reliability, and accuracy of 22 YouTube videos on exercises and management for dysphagia in pediatric populations. Of these videos, 36.4% and 72.7% did not have high quality or reliability, respectively. Half of the videos also did not contain accurate information. Even videos made by medical specialists lacked quality, reliability, and accuracy. As YouTube is one of the major sources of information, even on topics such as exercises and management for pediatric dysphagia, video makers, especially medical specialists, should make an effort to create videos with high quality, reliability, and accuracy of information.

## Figures and Tables

**Figure 1 children-09-01514-f001:**
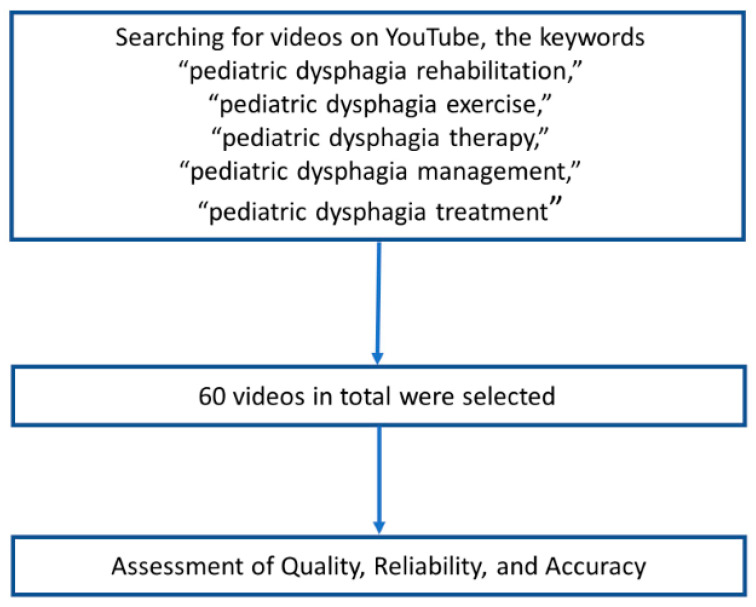
Flow diagram of this study.

**Table 1 children-09-01514-t001:** Global quality scale.

1. Poor quality, poor flow, most information missing, not helpful for patients
2. Generally poor, some information given but of limited use to patients
3. Moderate quality, some important information is adequately discussed
4. Good quality good flow, most relevant information is covered, useful for patients
5. Excellent quality and excellent flow, very useful for patients

**Table 2 children-09-01514-t002:** **Modified** DISCERN reliability tool.

1. Is the video clear, concise, and understandable?		
2. Are valid sources cited? (from valid studies, physiatrists, or physicians)		
3. Is the information provided balanced and unbiased?		
4. Are additional sources of information listed for patient reference?		
5. Does the video address areas of controversy/uncertainty?		

**Table 3 children-09-01514-t003:** General features of the videos.

Video Features	Mean ± SD (min–max)
Duration (s)	1230.73 ± 1932.78 (64–6807)
Number of views	47,902.45± 106,309.53 (95–379,736)
Number of likes	698.16 ± 2008.81 (1–8800)
Number of dislikes	0.00 ± 0.00 (0–0)
Number of comments	6.05 ± 15.13 (0–69)
GQS	3.68 ± 0.95 (2–5)
mDISCERN	1.77 ± 1.34 (0–5)

SD; standard deviations, GQS; global quality scale, mDISCERN; modified DISCERN.

**Table 4 children-09-01514-t004:** Categorization of the quality of videos according to sources, n (%).

Source	Low Quality	Intermediate Quality	High Quality	Total
Therapist	0 (0.0)	3 (33.3)	6 (66.7)	9 (40.8)
Physician or physiatrists	0 (0.0)	1 (25.0)	3 (75.0)	4 (18.2)
Health-related website	1 (100.0)	0 (0.0)	0 (0.0)	1 (4.6)
Academic	0 (0.0)	0 (0.0)	0 (0.0)	0 (0.0)
University/professional organization/association	0 (0.0)	1 (33.3)	2 (66.7)	3 (13.6)
Nonphysician health personnel	0 (0.0)	0 (0.0)	1 (100.0)	1 (4.6)
Patient	0 (0.0)	0 (0.0)	0 (0.0)	0 (0.0)
Independent user	1 (25.0)	1 (25.0)	2 (50.0)	4 (18.2)
Total	2 (9.1)	6 (27.3)	14 (63.6)	22 (100.0)

**Table 5 children-09-01514-t005:** Categorization of the accuracy of videos according to sources, n (%).

Source	Accurate	Mis-Leading	Total
Therapist	5 (55.6)	4 (44.4)	9 (40.8)
Physician or physiatrists	3 (75.0)	1 (25.0)	4 (18.2)
Health-related website	0 (0.0)	1 (100.0)	1 (4.6)
Academic	0 (0.0)	0 (0.0)	0 (0.0)
University/professional organization/association	2 (66.7)	1 (33.3)	3 (13.6)
Nonphysician health personnel	0 (0.0)	1 (100.0)	1 (4.6)
Patient	0 (0.0)	0 (0.0)	0 (0.0)
Independent user	1 (25.0)	3 (75.0)	4 (18.2)
Total	11 (50.0)	11 (50.0)	22 (100.0)

**Table 6 children-09-01514-t006:** Categorization of the reliability of videos according to sources, n (%).

Source	Reliable	Non-Reliable	Total
Therapist	3 (33.3)	6 (66.7)	9 (40.8)
Physician or physiatrists	1 (25.0)	3 (75.0)	4 (18.2)
Health-related website	0 (0.0)	1 (100.0)	1 (4.6)
Academic	0 (0.0)	0 (0.0)	0 (0.0)
University/professional organization/association	2 (66.7)	1 (33.3)	3 (13.6)
Nonphysician health personnel	0 (0.0)	1 (100.0)	1 (4.6)
Patient	0 (0.0)	0 (0.0)	0 (0.0)
Independent user	0 (0.0)	4 (100.0)	4 (18.2)
Total	6 (27.3)	16 (62.7)	22 (100.0)

**Table 7 children-09-01514-t007:** Correlations between quality, reliability, accuracy, and other video parameters.

	GQS	mDISCERN	Accuracy	View per Day	Like per Day	Comment per Day
GQS		0.766 *	0.738 *	0.466 *	0.311	0.501
mDISCERN			0.658 *	0.342	0.205	0.441
Accuracy				0.403	0.373	0.208
View per day					0.581 *	0.222
Like per day						0.824 *
Comment per day						

GQS; global quality scale, mDISCERN; modified DISCERN score. * *p* < 0.05.

**Table 8 children-09-01514-t008:** Multiple regression analysis among quality, reliability, and other various video parameters.

Dependent Variable	Independent Variables	R^2^	Beta Coefficient	Standard Error	Odd Ratio (95% CI)	*p* Value
GQS	mDISCERN	0.826	0.347	0.121	0.094~0.600	0.01 *
	Accuracy	0.826	0.763	0.317	0.100~1.427	0.026 *

GQS; global quality scale, mDISCERN; modified DISCERN score. * *p* < 0.05.

## Data Availability

The data presented in this study are available in Appendix A.

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
