# Peer review of "The Quality, Reliability, and Accuracy of Videos Regarding Exercises and Management for Dysphagia in Pediatric Populations Uploaded on YouTube"

_children, 2022, doi:10.3390/children9101514_

Round 1
Reviewer 1 Report
This is a well written manuscript about the accuracy of information on the treatment of dysphagia as presented in You Tube videos. I think it would be useful to the readers if the authors could prepare a table describing the type of common information errors made in the videos. This would help health care professionals who treat dysphagia to possibly make more accurate and useful videos for You Tube.
Author Response
Answer: We appreciate your valuable comment. We agree with your comment. Currently, there is no comprehensive and multidisciplinary recommendation study covering all aspects of pediatric dysphagia. Therefore, we used a recent study, which includes comprehensive and detailed answers for every problem that could be posed in clinical practice for the management of Pediatric Dysphagia, and recommendations are for all pediatric patients with both oropharyngeal and esophagpeal dysphagia, as a reference for judgement of the accuracy.
“Umay E, Eyigor S, Giray E, Karadag Saygi E, Karadag B, Durmus Kocaaslan N, Yuksel D, Demir AM, Tutar E, Tikiz C, Gurcay E, Unlu Z, Celik P, Unlu Akyuz E, Mengu G, Bengisu S, Alicura S, Unver N, Yekteusaklari N, Uz C, Cikili Uytun M, Bagcier F, Tarihci E, Akaltun MS, Ayranci Sucakli I, Cankurtaran D, Aykin Z, Aydin R, Nazli F: Pediatric dysphagia overview: best practice recommendation study by multidisciplinary experts. World J Pediatr, 2022.”
This previous study, using a seven-step process and a third-modified Delphi survey, provided comprehensive and detailed answers and recommendations to all issues that may arise in clinical practice for the management of dysphagia in children from the perspective of an experienced multidisciplinary expert. So, instead of tables, we once again emphasize the source of accurate information in the text. We believe that by emphasizing the information in this reference once again, readers will help authors obtain accurate information about current PD management. Thank you.
Reviewer 2 Report
Peer review report for Children, MDPI Journal
Article title: “The Quality, Reliability, and Accuracy of Videos Regarding Exercises and Management for Dysphagia in Pediatric Populations Uploaded on YOUTUBE”
Reviewers comments to the article:
The topic of the study is actual and interesting. Due to the possibility for everyone to immediately and freely access scientific information, experts should verify contents uploaded on web and social media. The aim of the study is great also because of the topic analyzed. Nonetheless, the manuscript in this actual form requires minor revisions due to methodological issues. The English language review should also be considered.
1. ABSTRACT: coherent with the article in the actual form.
2. INTRODUCTION: overall adequate.
3. MATERIALS AND METHODS: to allow correct interpretation and reproducibility of the results, in our opinion, all the sources used for the research (e.g. at least link to original videos) should be cited.
4. RESULTS: results of multivariate logistic regression are not adequately presented, in our opinion a revision by a statistic is required.
5. DISCUSSION: coherent with the actual form of the article, in our opinion the lack of videos uploaded by health organizations and/or medical associations (previously reported to be the most reliable sources, Madathil et al. 2015) should be stressed.
6. CONCLUSIONS: coherent with the actual form of the article.
7. REFERENCES: coherent with the actual form of the article.
8. TABLES: see table 3: are mean and SD in row 3,4 and 5 correct?
The authors of this peer-review declare that they have no conflict of interest.
Author Response
Reviewers comments to the article:
The topic of the study is actual and interesting. Due to the possibility for everyone to immediately and freely access scientific information, experts should verify contents uploaded on web and social media. The aim of the study is great also because of the topic analyzed. Nonetheless, the manuscript in this actual form requires minor revisions due to methodological issues. The English language review should also be considered.
- ABSTRACT: coherent with the article in the actual form.
Answer: We appreciate your valuable comment.
- INTRODUCTION: overall adequate.
Answer: We appreciate your valuable comment.
- MATERIALS AND METHODS: to allow correct interpretation and reproducibility of the results, in our opinion, all the sources used for the research (e.g. at least link to original videos) should be cited.
Answer: We appreciate your valuable comment. Following your comment, we have added a link of YouTube video as a supplementary files.
- RESULTS: results of multivariate logistic regression are not adequately presented, in our opinion a revision by a statistic is required.
Answer: We appreciate your valuable comment. Following your comment, we have modified the method of our statistical analysis.
- DISCUSSION: coherent with the actual form of the article, in our opinion the lack of videos uploaded by health organizations and/or medical associations (previously reported to be the most reliable sources, Madathil et al. 2015) should be stressed.
Answer: We appreciate your valuable comment. Following your comment, we have added it in discussion section.
- CONCLUSIONS: coherent with the actual form of the article.
Answer: We appreciate your valuable comment.
- REFERENCES: coherent with the actual form of the article.
Answer: We appreciate your valuable comment.
- TABLES: see table 3: are mean and SD in row 3,4 and 5 correct?
Answer: We appreciate your valuable comment. We checked again .Thank you.

Reviewer 3 Report
I think this study is very interesting and a major eye opener for the reality of what is on youtube and most likely being used with a negative impact.
No major concerns. I think the manuscript needs to be revised for grammar. Many of the tenses are incorrect.
YouTube instead of YOUTUBE needs to be consistent throughout the manuscript.
Many sentences have duplicated or incorrectly placed words that hinders the clarity of some sentences.
Also the references must go before the punctuation.
When those are fixed I think this is a solid manuscript.
Author Response
Answer: We appreciate your valuable comment. Following your comment, we have modified it. Thank you.